

# Investigations of Temporal and Spatial Distribution of Precursors SO$_2$ and NO$_2$ Vertical Columns in North China Plain by Mobile DOAS

Fengcheng Wu[1], Pinhua Xie[1,3,4*], Ang Li[1], Fusheng Mou[1], Hao Chen[1], Yi Zhu[2], Tong Zhu[2], Jianguo Liu[1], Wenqing Liu[1]

[1] Key Laboratory of Environmental Optical and Technology, Anhui Institute of Optics and Fine Mechanics, Chinese Academy of Sciences, Hefei, 230031, China

[2] State Key Laboratory of Environmental Simulation and Pollution Control, College of Environmental Sciences and Engineering, Peking University, Beijing, 100871, China

[3] Center for Excellence in Regional Atmospheric Environment, Institute of Urban Environment, Chinese Academy of Sciences, Xiamen, 361021, China

[4] School of Environmental Science and Optoelectronic Technology, University of Science and Technology of China, Hefei, 230026, China

*Correspondence to:* P. H. Xie (phxie@aiofm.ac.cn)

**Abstract:** Recently, Chinese cities have suffered severe events of haze air pollution, particularly in the North China Plain (NCP). Investigating the temporal and spatial distribution of pollutants, emissions, and pollution transport is necessary to better understand the effect of various sources on air quality. We report on mobile differential optical absorption spectroscopy (mobile DOAS) observations of precursors SO$_2$ and NO$_2$ vertical columns in NCP in summer of 2013 (from 11 June to 7 July) in this study. The different temporal and spatial distributions of SO$_2$ and NO$_2$ vertical column density (VCD) over this area are characterized under various wind fields. The results show that the transport from southern NCP strongly affects the air quality in Beijing, and the transport route, particularly SO$_2$ transport of Shijiazhuang–Baoding–Beijing is identified. In addition, the major contributors to SO$_2$ along the route of Shijiazhuang–Baoding–Beijing are elevated sources and low area sources for the route of Dezhou–Cangzhou–Tianjin–Beijing are found using the interrelated analysis between in situ and mobile DOAS observations during the measurement periods. Furthermore, the discussion of hot spot near Ji'nan City shows that the average observed width of polluted air mass is 11.83 km and 17.23



km associated with air mass diffusion, which is approximately 60 km away from emission sources based on geometrical estimation. Finally, a reasonable agreement exists between OMI and mobile DOAS observations with correlation coefficient ($R^2$) of 0.65 for $NO_2$ VCDs. Both datasets also have similar spatial pattern. The fitted slop of 0.55 is significantly less than unity can reflect the contamination of local sources and OMI observations need to improve the sensitivities to the near-surface emission sources through the improvements of retrieval algorithm or resolution of satellites.

## 1. Introduction

Driven by the unprecedented economic growth and explosive increase in urbanization, China has been experiencing severe air pollution, particularly in developed areas, such as, Yangtze River Delta region and Pearl River Delta region (van Donkelaar et al., 2010). The severe haze pollution events occurred frequently since the end of 2012 in Jing-Jin-Ji region, including Beijing, Tianjin, Shijiazhuang, and some cities in Hebei province. Long duration, heavy pollution level, and large spread area are the main characteristics of haze pollution, which have been rare in the past decades (Sun et al., 2014; Ji et al., 2014; Zhao et al., 2013). Haze pollution has affected the health and lifestyle of millions, drawing extensive worldwide attention on China. Severe air pollution in Beijing, the capital of China, has troubled the public, scholars, and the government. Concurrently, many studies have been conducted in Beijing and its surrounding areas (Wang et al., 2014a; Wang et al., 2014b; Xu et al., 2011; Ma et al., 2012). Related results show that the air pollution in Beijing is a regional environmental problem caused by the influences of both local emission and external transport (Ying et al., 2014; Guo et al., 2014; Wu et al., 2011).

$NO_2$ is one of the most important atmospheric trace gases. It plays a key role in tropospheric and stratospheric chemistry and strongly participates in the chain reaction formation of tropospheric ozone (Crutzen et al., 1970). Moreover, $NO_2$ is the main pathway of OH loss, which determines the atmospheric oxidative capacity, under heavy polluted conditions (Finlayson-Pitts et al., 1999; Kanaya et al., 2014). Aside from $NO_2$ being generally harmful to human health, long-term $NO_2$ exposure in high concentrations can also increase the possibility of bronchitis in asthmatic children (WHO, 2006). Combustion processes, such as power generation and release of pollutants from vehicles, are the major sources of anthropogenic $NO_2$ emissions. Meanwhile, $SO_2$ is a colorless gas that adversely affects the





respiratory system. Emissions from elevated releases, such as that from power plants, are the main

contributor for anthropogenic $SO_2$ emission (Xu et al., 1998; Ramanathan et al., 2003). Furthermore,

$NO_2$ and $SO_2$ are important precursors of aerosol. Under suitable meteorological conditions, $NO_2$ and

$SO_2$ tend to form nitrate and sulfate, which contribute to the formation of secondary aerosols (Jang et

al., 2001; Boichu et al., 2015). Some studies show that nitrate and sulfate account for large proportions

in particulate matter ($PM_{2.5}$, with aerodynamic diameter less than or equal to 2.5μm), which is an

important element of haze, in Jing-Jin-Ji region (Huang et al., 2014; Yang et al., 2011, Sun et al., 2013,

Zhang et al., 2013). Based on model simulation, $PM_{2.5}$ concentration can be significantly reduced if

$SO_2$ and $NO_x$ emission have been controlled effectively (Zhao et al., 2013). In addition, the spatial and

temporal distributions of $SO_2$ and $NO_x$ vary significantly (Lee et al., 2009;Matsui et al., 2009; Wang et

al., 2009). To investigate the spatial and temporal distribution of $SO_2$ and $NO_2$ and evaluate the

influence of transport on Beijing the observations of distribution of tropospheric $SO_2$ and $NO_2$ vertical

column density (VCD) was conducted in North China Plain (NCP) using mobile DOAS from June to

July 2013. NCP is located in northern China, surrounded by Taihang Mountain (at the west of NCP),

Yanshan Mountain (at the north of NCP) and Bohai Sea (at the east of NCP). NCP consists of the

Jing-Jin-Ji region and other provinces in Northern China and is one of the heaviest polluted areas in

China (Quan et al., 2011).

A large number of studies on distributions of air pollutants have also been performed in NCP. The

characteristics of concentration and evolution at different sites and formation mechanisms during heavy

pollution periods have been researched using ground-based observation networks (Hu et al., 2014).

Meanwhile, regional variations of gas, particle pollutants, and other factors which influence pollution

characteristics have been detected using airborne measurement (Zhang et al., 2014). Also, based on

mobile laboratory, Wang et al. (2011) analyzed the regional distribution of $SO_2$ in Beijing and its

surrounding areas and estimated transport flux from the outside to Beijing (Wang et al., 2011).

Simulation model, another alternative method, can obtain distribution, transboundary transport fluxes,

and major transport channel of Beijing in combination with meteorological data (An et al., 2012).

However, current studies mainly focus on ground-based observations, lacking stereoscopic monitoring

data that can help better understand the source and transport of air pollution.

Mobile DOAS provides another novel remote sensing method to obtain stereoscopic monitoring data

and characterize regional distribution of air pollution over medium- to large-distance scale. This




technique can detect the horizontal distribution of pollutants with high spatial–temporal resolution and rapidly identify the location of pollution sources. Furthermore, information on the upper layer of air pollution can be investigated. Thus, transport of air pollution can be analyzed and associated with meteorological trajectory data. At present, some related studies have been carried out (Ibrahim et al.,

2010; Shaiganfar et al., 2011). In China, several measurements are also performed in Shanghai and Guangzhou. Wang et al. (2012) evaluated the $NO_2$ variation over the central urban area before and after Shanghai Expo 2010 (Wang et al., 2012). Wu et al. (2013) observed the distribution and emission of $SO_2$ and $NO_2$ in Guangzhou Eastern Area during Guangzhou Asian Games 2010(Wu et al., 2013). However, this study is to summarize the distribution of $SO_2$ and $NO_2$, verify the type of air pollution

sources, and to discuss potential of transport from NCP to Beijing over NCP area. In addition, the mobile platform referred in this study is also equipped with some point instruments from Peking University (PKU) to synchronously measure near-surface concentration of gas and particulate mass. In this paper, we present the observations of $SO_2$ and $NO_2$VCD in NCP from June to July 2013 using mobile DOAS, and the distributions of $SO_2$ and $NO_2$ VCD in NCP are characterized. In combination

with ground surface data (point instrument), the characteristics of $SO_2$ and $NO_2$ along southwest and southeast pathway under different wind fields are characterized and the hot spots and their possible sources along the measurement paths are determined. The pollution transport pathways to Beijing are revealed and first time convinced by capturing the plume. Finally, the $NO_2$ VCDs from mobile DOAS data are compared with those from Ozone Monitoring Instrument (OMI). Obtained data are in good

agreement.

This paper is organized as follows: the experimental process, including overview of the measurements and instruments. Wind fields and the principle of retrieval of vertical density of tropospheric trace gas are discussed in detail in Sect. 2. Section 3 gives us the results and discussions, including distributions of $SO_2$ and $NO_2$ tropospheric VCDs over NCP and analysis of hot spot and comparison with OMI $NO_2$.

Finally, the conclusions are presented in Sect. 4.

## 2.   Experimental

### 2.1 Overview of the measurements

To characterize the spatial distributions of $SO_2$ and $NO_2$ VCD and investigate the potential transport to

Beijing, the measurement routes are specially designed. The entire measurement period from 11June to



16 July 2013 is initially divided into five identical cycles. Mobile DOAS observations span four cycles (from 11 June to 7 July) and cover four different routes because of bad weather or vehicle problem. Figure 1 depicts the detailed routes of mobile DOAS measurements. Cycle 1 covers five routes with a total path of 1400 km and takes five days to complete. The five routes are Beijing (BJ) to Shijiazhuang

(SJZ), Shijiazhuang to Dezhou (DZ), Dezhou to Baoding (BD) to Cangzhou (CZ), Cangzhou to Zhuozhou (ZZ), and Zhuozhou to Beijing for cycle1. Due to the bad weather or vehicle problem, Cycles 2 and 3 took four and three days to complete, resulting in skipping of some routes. We needed one more day to complete Cycle 4 due to power failure on 4 July. The details of monitoring information are listed in Table 1.

The approximate starting and ending times are 10:00 and 14:00 (Local Time, LT), particularly considering stable boundary layer and battery endurance. During the entire measurement period, the temperature varied from 30 ℃ to 36 ℃, and the wind fields were dominated by south and north.

**2.2 Instrument description**

Mobile DOAS instrument collects scattered sunlight from zenith observation. Details of the instrument

and performances are described in our previous study (Wu et al., 2013). Briefly, the system consists of telescope, a miniature fiber spectrometer unit, global positioning system, and computer. The series of Ocean Optics HR2000 is selected as miniature spectrometer, with spectral resolution of 0.6 nm and spectral range of 290 nm to 420 nm. The spectrometer is stored in a temperature-controlled unit to stabilize the temperature at +30±0.1 ℃ to avoid spectral shifts caused by temperature variations. The

detection limits of the instrument is approximately $3–5 \times 10^{15}$ molec./cm$^2$ for $SO_2$ and $NO_2$. The instrument is installed on an IVECO Turin V diesel vehicle (L = 6.6 m, W = 2.4 m, H = 2.8 m; payload = 2.7 metric tons), which is a mobile laboratory platform from PKU (Wang et al., 2009, 2011). The mobile DOAS system is powered by 220V alternating current through conversion of +12V direct current battery with a power converter.

In addition, PKU has set up some point instruments on the van, including $SO_2$ analyzer (ECOTECH 9850A), $NO_x$ analyzer (ECOTECH 9841A), CO analyzer (ECOTECH 9830A), ozone analyzer (ECOTECH 9810A), and $CO_2$ analyzer (ECOTECH 9820A). Aside from gaseous pollutant instruments, some aerosol instruments, such as GRIMM and Dusttrak for $PM_{2.5}$ and Fast Mobility Particle Sizer, were also available onboard for analysis of particle size distribution. The details of the setup and

performance of the instruments are described in Wang et al. (2009, 2011).





### 2.3 Backward Lagrangian Trajectory Simulation

Apart from the near-surface wind data, the backward trajectory of air mass from the stations in Beijing was also simulated using the Hybrid Single Particle Lagrangian Integrated Trajectory model (HYSPLIT, offline version), which has been developed by the Air Resources Laboratory of the US National

Oceanic and Atmospheric Administration. An average boundary layer height (BLH) of around 1000m has been calculated during noontime in summer over NCP area by Lv et al based on lidar observations. The middle altitude of BLH, i.e., 500m, is taken as the representative horizontal transport height to investigate transport effect on an assumption of well mixing throughout the whole BL around noontime. So, backward trajectories were calculated once every 2 h for 1 day (24 h) at a selected altitude of 500 m

above ground level for each cycle. An archive meteorological database with a horizontal resolution of 1 ° × 1 ° from the Global Data Assimilation System, which is enough to identify the original regions of air mass, is chosen to run the HYSPLIT model.

Figure 2 shows the cluster average backward trajectory for Cycles 1, 2, 3, and 4. During the measurement periods of Cycles 1 and 3, all air masses came from the southern regions. For Cycle 2, the

mean back trajectory is roughly split equally between north and south. However, the dominant wind field is north during the mobile DOAS observations for Cycle 2, except for the wind on 21 June as listed in Table 1. A small different wind field is present for Cycle 4. The north wind account for nearly 72% and the south wind 28%. The detailed different observation periods of mobile DOAS exhibit that dominant wind is south except for that on 2 and 5 July.

### 2.4 Retrieval of Vertical Density of Tropospheric Trace Gas

$SO_2$ and $NO_2$ column densities are retrieved from zenith sky mobile DOAS with WinDOAS software. Each measured spectrum starts with dark current, and offset corrections is divided through Fraunhofer reference spectrum (FRS), which is a relative "clean-air" spectrum. One FRS spectrum is selected to retrieve all other measured spectra during the whole field campaign. The FRS is recorded at

approximately 11:30 LT on April 30, 2013, near Bohai Sea, considering strong wind and good air quality on that day (see Fig. 1a). After the retrieval procedures of logarithm of the ratio of measured spectrum to FRS spectrum and high-pass and low-pass filtering, the differential slant column density (DSCD), which is relative to the value of Frauenhofer spectrum, is obtained. A retrieval example of $SO_2$ and $NO_2$ DSCD are illustrated in Fig. 3.

The wavelength range of $SO_2$ fit is 310 nm–324 nm and 338 nm–379 nm for $NO_2$ fit with WinDOAS.



The high-resolution absorption cross-sections of $SO_2$, $NO_2$, HCHO, and $O_3$ at 293 K from Bogumil et al. (2003) are used in DOAS $SO_2$ fit. The $NO_2$ fit included the cross-section of $NO_2$, HCHO, and $O_3$ at 293K (Bogumil et al., 2003) and $O_4$ at 298 K (Greenblatt et al., 1990). In addition, the FRS and Ring spectrum are also included. The synthetic Ring spectrum is yielded from FRS spectrum using DOASIS

software (Kraus, 2006). The slit function is generated from the emission peak of mercury lamp at 334 nm. The high-resolution solar spectrum is used to calibrate wavelength. The fit uncertainties of $NO_2$ and $SO_2$ for the spectrum, as shown in Fig. 3, are approximately 2.48% and 1.84%. The typical uncertainties are less than 15% for $NO_2$ and 20% for $SO_2$.

The above obtained the DSCD with respect to FRS spectrum. Tropospheric $NO_2$ is $\sim 5 \times 10^{15}$ molec./cm$^2$

at the location of FRS spectrum on 30 April, 2013 from OMI result. Given the poor $SO_2$ satellite data, we checked the $SO_2$ results at ground level from a local environmental protection agency on that day. Compared with the high pollution over NCP area, we neglected these relatively small tropospheric contents in FRS spectrum. As a result, the tropospheric $NO_2$ and $SO_2$ VCD can be calculated with air mass factor (AMF) using the following formula:

$$VCD_{trop} = \frac{SCD_{trop}}{AMF_{trop}} = \frac{DSCD + SCD_{FRS} - SCD_{strat}}{DAMF + AMF_{FRS} - AMF_{strat}} = \frac{DSCD}{AMF_{trop}} \quad (1)$$

The radiative transfer model McArtim (Deutschmann et al., 2011) based on the Monte Carlo method is used to calculate the $AMF_{trop}$. We assume that aerosol and trace gas profiles are homogeneous below the BLH, whereas exponential profiles are above. Here, the constant concentrations within 1000m of boundary layer are assumed to be approximate 40 ppb and 10 ppb for $NO_2$ and $SO_2$ according to

state-controlled air-sampling sites. This hypothesis can lead to less than 5% uncertainty based on a sensitivity study by varying the setting of $NO_2$ and $SO_2$ setting. The average aerosol optical density (AOD) of 1.0 is estimated from the AERONET on June and July, 2013 at Xianghe site. The profiles of aerosol, $NO_2$, and $SO_2$ are taken from the LOWTRAN database and US standard atmosphere above the boundary layer. We estimate the total retrieval errors of $NO_2$ VCD and $SO_2$ VCD to be less than 20%

and 25% (Wu et al., 2013).

3.   **Results and discussion**

**3.1  Distributions of $SO_2$ and $NO_2$ Tropospheric VCDs over NCP**



In this section, the distributions of $SO_2$ and $NO_2$ tropospheric VCDs over NCP area are discussed with mobile DOAS observations. First, the overall distributions of $SO_2$ and $NO_2$ tropospheric VCDs along the measurement routes under the different dominant winds are characterized. Furthermore, we analyze the spatial and temporal variations of $SO_2$ and $NO_2$ tropospheric VCDs along the southwest routes (Shijiazhuang–Baoding–Beijing) and southeast routes (Dezhou–Cangzhou–Tianjin–Beijing) for different wind fields. The possible transport route of trace gas is identified using these distribution characteristics.

### 3.1.1 Overall Distributions of $SO_2$ and $NO_2$ Tropospheric VCDs

Completing each cycle measurements takes four to five days, and this can lead to the probed air mass change when meteorological condition varies rapidly. However, as described in Sect. 2.3 and listed in Table 1, the dominant wind field as a main influencing factor on air mass variation has not significantly changed, particularly the dominant wind direction of southerly and northerly winds during the measurement periods for mobile DOAS. However, air mass variation can also be affected by some other factors, such as temperature, humidity, and pressure, but the atmospheric physical reaction processes is too complicated to discuss in this study. Thus, we assume that, in this work, the air mass does not change dramatically for each cycle measurements.

Typical spatial distributions of $SO_2$ and $NO_2$ along the measurement route over NCP area for north and south wind fields are shown in Fig. 4. The maps of $SO_2$ in Fig. 4 show that increased values are observed under the southerly wind, particularly the results along Taihang Mountain, which is also part of the southwest measurement route (Shijiazhuang-Baoding-Beijing). The high $SO_2$ VCDs detected in the region near the cities of Shijiazhuang, Baoding, and Beijing indicate that these regions have emission sources of $SO_2$. In addition, high $SO_2$ VCDs are also observed on the cross-section of south route, particularly near Ji'nan City. This hot spot can always be found under the southerly wind during the field campaign, suggesting a strong emission outside the measurement area and south of it. Based on the backward trajectory analysis, the big air pollution plume comes from Liaocheng City, which is another small city close to Ji'nan western region. Furthermore, relatively low $SO_2$ VCDs are observed along the southeast route compared with that of the southwest route.

However, for the northerly wind, no increased $SO_2$ VCDs are noted along the Taihang Mountain. The hot spot near Ji'nan City also disappeared. The downwind $SO_2$ VCDs of Shijiazhuang and Tianjin city are relatively high due to source emission near the city. The results of comparison of wind direction



from south versus north further suggest that the strong emission sources located at the southern region of the measurement area have a significant influence on Beijing under the southerly wind, particularly along the Taihang Mountain.

Unlike $SO_2$, no significant difference between the southerly and northerly wind for $NO_2$ VCDs was noted. The $NO_2$ VCDs are affected by the "city effect." High $NO_2$ VCDs are obtained near Beijing and Shijiazhuang City. The same is noted for $SO_2$, and due to strong emission source contribution, enhanced $NO_2$ VCDs are also found near Jinan City.

### 3.1.2 Spatial and temporal variations of $SO_2$ and $NO_2$ along southwest and southeast routes under different wind fields

As detailed in above analysis, the characteristics of $SO_2$ and $NO_2$ distributions have significant variations, including spatial and temporal differences along the southwest and southeast measurement routes. This section firstly investigates the $SO_2$ and $NO_2$ characteristics along southwest and southeast routes and then compares them with the results under southerly and northerly wind.

Table 2 lists the $SO_2$ and $NO_2$VCD from mobile DOAS and near-surface concentration from point instrument under southerly and northerly wind along the southwest and southeast routes. For the southwest measurement route, the mean VCDs of $SO_2$ and $NO_2$ are $4.22 \times 10^{16}$molec./cm$^2$ and $1.69 \times 10^{16}$molec./cm$^2$. The mean near-surface concentration is 9.74 ppb and 111.28 ppb for $SO_2$ and $NO_2$. For the southeast measurement route, the mean VCDs of $SO_2$ and $NO_2$ are $3.40 \times 10^{16}$molec./cm$^2$ and $1.15 \times 10^{16}$molec./cm$^2$. The mean near-surface concentration of $SO_2$ and $NO_2$ is 17.27 ppb and 117.97 ppb.

The VCDs along southwest route are higher than that along the southeast route. However, the near-surface concentration along the different routes is reverse.

The vertical column and in situ measurements are discussed simultaneously in Table 2. It is interesting to note that such discussions can provide comprehensive information about surface emission and tropospheric pollution. We can also calculate the depth of a layer of air with the in situ mixing ratio and the measured vertical column on the assumption of homogenous mixing within the planetary boundary layer (Chen et al., 2009). However, the height of the layer could not be estimated in this way in this study because the in situ measurements contaminated by very local vehicle emission, especially for $NO_2$. The traffic exhaust is one of the major contributors to $NO_2$ and large traffic emission result in the inhomogenous mixing within the planetary boundary layer, so it is found that the $NO_2$ layer would be something like only 30 to 60m thick using above analysis method, which is very unreasonable





compared to normal situations. In contrast, the $SO_2$ layer would vary from about 0.5 to 2.0 km thick, which is in the normal range.

The comparisons of VCDs between different wind fields show that the VCDs under southerly wind are much higher than that under northerly wind along the southwest route, particularly for $SO_2$, with the

value of $6.09 \times 10^{16}$ molec./cm$^2$ and $2.35 \times 10^{16}$ molec./cm$^2$. However, this phenomenon is not significant along the southeast route. In addition, the comparisons of $SO_2$ near-surface concentration suggest that the difference between the different wind fields is not significant along the southwest route, but is enhanced dramatically along the southeast route under southerly wind, with the value of 23.29 ppb versus 11.24 ppb under the northerly wind.

**3.1.3 Characterization of emission sources and identification of transport route**

Both results from mobile DOAS and point instruments observations for every measurement day are shown in Fig. 5 to Fig. 8. According to the "Box-Chart" plot, some distinct peak values of $SO_2$ VCDs are measured in the case of the south wind, whereas this is not significant for $SO_2$ near-surface concentration, as shown in Fig. 5. These findings indicate that the elevated sources existed along the

southwest measurement route, and the elevated sources are the main $SO_2$ sources along the southwest route. We could also infer that the high $SO_2$ value may be located in the upper layer. No significant peak values for $NO_2$ VCD (see Fig. 6) are noted. However, we found them on near-surface concentration, such as on 11 June. The results show that the traffic emission located at the near surface is the main sources of $NO_2$. If we traverse areas with large volumes of vehicles, the $NO_2$ near-surface

concentration should increase. However, for the southeast measurement routes, we did not observe the peak values of $SO_2$ VCDs and near-surface concentrations, as shown in Fig. 7. One interesting finding is that the $SO_2$ VCD on 21 June increased slightly from "Box-Chart" plot in Fig. 7. These findings verify that the low nonpoint sources are the main contributors along the southeast routes. However, the emissions of elevated sources from western part of the measurement region could account for the $SO_2$

peak value on 21 June (the dominant wind is west on that day), and this is in agreement with the results along southwest routes. As shown in Fig. 8, the $NO_2$ VCDs for south wind are 1.38 times higher than that for north wind, but the near-surface concentrations are almost equal for these two different winds. The same is true for $SO_2$; we also did not find elevated $NO_2$ sources along the southeast routes.

Based on the above analysis, we could infer that the pollution source along the southwest and southeast

routes have two types. The finding is also proven by the emission inventory: several large emission



sources are located southwestern region, and some near-surface fugitive sources are located southeastern region (Wang et al., 2011).

Similar with $SO_2$, the average $NO_2$ VCD along the southeast route is lower than that along the southwest route, but the near-surface concentration is higher than that along the southwest routes. The

near-surface vehicle emissions are the major contributors of $NO_2$, and fugitive emission sources are additional sources of $NO_2$. In addition, the high $NO_2$ near-surface concentration along the southeast route indicates large traffic volume over this region. This is also consistent with the fact that the southeast route is an expressway from Beijing to Shanghai, the two most economically developed cities in China. Additionally, trade exchanges among these two and other cities are frequent.

The VCDs and near-surface concentrations of $SO_2$ and $NO_2$ are high under the southerly wind in most cases, particularly for $SO_2$ VCDs along the southwest routes and $SO_2$ near-surface concentrations along the southeast routes. From mobile DOAS observations, the significant variations of $SO_2$ VCDs along the southwest routes (also along Taihang Mountain) are shown in Fig. 9. The variations of $SO_2$ VCDs for the different wind fields indicate that the southwest route is a transport route of $SO_2$ for Beijing.

When the air plume comes from the south, the air quality in Beijing deteriorates. Figure 10 shows the mean $SO_2$ concentrations for when south or north wind is dominant in Beijing. The monitoring data in seven state-controlled air-sampling sites demonstrate that the average $SO_2$ concentrations ranged from 8.22 ppb to 13.04 ppb for south wind and from 3.71 ppb to 5.02 ppb for north wind during the mobile DOAS observation period in the Beijing area. Previous studies also confirmed the presence of this

transport route using other methods (Su et al., 2014). This work not only identifies the transport route of $SO_2$ with mobile DOAS observations, but also determines the high $SO_2$ concentration existing in the upper layer combination the concurrent near-surface data.

### 3.2 Analysis of hot spot

The hot spots are observed for the route of Shijiazhuang–Dezhou measurements under southerly wind.

The maximum $SO_2$ VCD and $NO_2$ VCD can reach $4.84 \times 10^{17}$ molec./cm$^2$ and $7.41 \times 10^{16}$ molec./cm$^2$. However, the hot spots are not detected for the north wind. Figures 11 and 12 present the results of $SO_2$ and $NO_2$VCDs for the Shijiazhuang–Dezhou measurements under southerly and northerly wind.

Figure 11 exhibits a large polluted air mass coming from southern region in the rectangular area [Fig. 11 (a1) and Fig. 11(b1)] under southerly wind on 12 June. First, this air mass led to the rapid

enhancement of $SO_2$ and $NO_2$ VCDs in Area I and Location 1 [Fig.11 (a1), (a2), (b1), and (b2)]; then,



the VCDs in Area II and Location 2 increased subsequently due to the air mass diffusion. The time

series of SO$_2$ and NO$_2$ VCDs tell us that with the increase of distance, the peak value decreased in Area

II and observed width of air mass enlarged because of air mass diffusion. For Area I, the peak values of

SO$_2$ and NO$_2$ VCDs are $4.43 \times 10^{17}$ molec./cm$^2$ and $6.80 \times 10^{16}$ molec./cm$^2$ at 13:02 (LT). However, the

peak value for SO$_2$ and NO$_2$ decreased to $3.44 \times 10^{17}$ molec./cm$^2$ and $4.68 \times 10^{16}$ molec./cm$^2$ in Area II

at 13:43 (LT).

Furthermore, observed widths of air mass are estimated in Area I and Area II from the time series of

SO$_2$ VCDs in Fig. 10 (a3) using the following formula:

$$W = \sum_{i} (t_{i+1} - t_i) \cdot \overline{V}_{i+1 \to i} \quad (2),$$

where $i$ is the number of spectrum in Fig. 11(a3), $t_{i+1}$ and $t_i$ are the time for the spectrum of $i$ and

$i + 1$, and $\overline{V}_{i+1 \to i}$ is the mean car speed between $t_{i+1}$ and $t_i$.

Using the above formula, the average observed width of air mass is calculated to be 11.83 km in Area I

and 17.23 km in Area II. Combined with the observed widths for Areas I and II and the geometric

relationships between these two locations, the distance of the air pollution sources from the Area I is

estimated at a distance of approximately 61.39 km. The distance of Area I from Liaocheng City is

approximately 60 km, proving that the source is indeed located in Liaocheng City, as discussed in Sect.

3.1.1.

While the above peak values are not found under the northerly wind as shown in Fig. 12, this

phenomenon further confirmed the large sources located at the southern region outside the

measurement area. When the dominant wind comes from south, the air quality of the measurement area

is severely influenced by the sources.

In addition, we simultaneously compare the one-minute average VCD results with the one-minute

near-surface concentrations along the Shijiazhuang–Dezhou routes. Figure 13 shows the time series of

VCDs and near-surface concentrations for SO$_2$ and NO$_2$ along the measurement route under the

southerly and northerly wind. For the specific southerly wind, such as on 3 July, the high SO$_2$ and NO$_2$

VCDs were captured through mobile DOAS in the areas, as shown in Fig. 11 (a2) or (b2) (the area

marked with sparse rectangular box in Fig. 13). This also indicates that the polluted air mass contained

high levels of SO$_2$ and NO$_2$. Furthermore, from the time series observations of SO$_2$ near-surface

concentrations, high near-surface concentrations are observed simultaneously in the sparse rectangle, as



shown in Fig. 13 (a), and this is the same as $SO_2$ VCDs. The combined results demonstrate that part of the air mass have deposited, resulting in the increase of $SO_2$ near-surface concentration. However, one interesting thing has been found the $NO_2$ near-surface concentration does not significantly increase in this area [Fig. 13 (b)]. Following the above explanation regarding $SO_2$, the declined air mass is

supposed to cause an increase in $NO_2$ near-surface concentration. The lifetime of $NO_2$ is less than $SO_2$, and the $NO_2$ conversion to other species, such as nitrate, could account for this unexpected finding. For the northerly wind, both VCDs and near-surface concentrations do not increase in the sparse box area. The correlation analysis between $NO_2$ and CO near-surface concentrations [Fig. 13 (e) and (f)] shows that $NO_2$ near-surface concentration mainly results from vehicle exhaust, regardless of the specific

wind direction.

### 3.3 Comparison with OMI $NO_2$

The OMI, which is onboard the Aura satellite of the Earth Observing System, was launched on 15 June 2004 with a nadir viewing mode (Levelt et al., 2006). OMI can be used to monitor global atmospheric $NO_2$ distribution with high spatial (up to $13 \times 24$ km) and temporal (daily global coverage) resolution.

OMI is equipped with two charge-coupled devices spanning a wavelength range from 264 nm to 504 nm to measure the solar scattering in the ultraviolet and visible spectra. The overpass time of OMI is 13:45 (LT) on the ascending node. In the current study, the OMI tropospheric $NO_2$ data product from NASA is used. The data consisted of three steps to derive tropospheric $NO_2$ VCD. The SCDs of $NO_2$ are derived from the OMI collected spectra based on the DOAS method in the wavelength ranging

from 405 nm to 465 nm. The air mass factors (AMFs) are applied to convert the SCDs to VCDs and are calculated with the monthly average $NO_2$ profile shapes (Rotman et al., 2001). Finally, the stratospheric contribution is estimated to derive tropospheric $NO_2$ VCDs by subtracting the stratospheric columns. Detailed description of tropospheric $NO_2$ retrieval process can be found in Bucsela et al. (2013).

In this study, to achieve a better comparison between OMI and mobile DOAS, the OMI tropospheric

$NO_2$ data are gridded onto a $0.1\,° \times 0.1\,°$ with an error and area weighted gridding algorithm (Wenig et al., 2008). The cloud fraction of 0.4 is used as a threshold to filter out the data with a cloud cover larger than 0.4. As a result, a total of 8 days (11, 12, 13, 18, 25, and 26 June and 3 and 6 July) of measurements from both OMI and mobile DOAS can be used for data comparison.

The comparisons of $NO_2$ VCDs between both datasets for our 8-day measurement are shown in Fig. 14.

The similar spatial pattern is captured by both OMI and mobile DOAS. In most cases, high level of





NO$_2$ VCD is observed around Shijiazhuang area. However, the hot spots of mobile DOAS observations, as shown in Sect. 3.2, cannot be detected completely using OMI due to the insensitivity of OMI observation to near-surface sources.

However, the NO$_2$ VCDs along southwest route are higher than that along the southeast route from

OMI observations, the same as discussed in Sect. 3.1.2. Moreover, the mobile DOAS data are averaged within each gridded satellite pixel (0.1 °$\times$ 0.1 °) and compared to OMI values within each gridded pixel for every day of the 8-day measurement. And the correlation analysis for all the datasets of the 8-day measurement is shown in Fig. 15. The correlation coefficient (R$^2$) is 0.65, suggesting that both observations agree reasonably well. However, a systematic difference between the mobile DOAS and

OMI NO$_2$ VCDs is exist as shown in Fig. 15 implied by the fitted slope that is significantly less than unity (0.55). These discrepancies can be attributed to source emission from near surface (e.g. traffic exhaust, industrial sources etc.) or lower troposphere (e.g. elevated sources). The OMI observations are insensitivity to the local sources due to the limited spatial resolution and shield by aerosols and clouds. However, the mobile DOAS have an ability of rapid response to local sources, especially for lower

troposphere, like elevated sources. Also some studies have shown that the OMI NO$_2$ VCDs are systematically smaller than mobile DOAS and MAX-DOAS observations over polluted areas (Shaiganfar et al., 2011, Chan et al., 2015). Of course, some other factors can also result in these differences, like NO$_2$ diurnal cycle, different hypothesis on calculation of AMFs etc. The measurement time of mobile DOAS is approximately from 10:00 to 14:00, and the OMI overpass time may be 13:45.

Previous studies (Wu et al., 2013) have shown that the strong diurnal variations of NO$_2$ occur between 10:00 and 11:00. As a result, the time mismatch between OMI and NO$_2$ could result in different NO$_2$ VCDs. In addition, the NO$_2$ VCDs of OMI and mobile DOAS need to be converted from SCDs with AMFs. The calculations of AMFs should consider the trace gas profiles, aerosol profiles, ground albedo and wavelength etc. These different sets can yield different AMFs and VCDs.

### 4.  Conclusions

The NCP has been experiencing severe air pollution associated with unprecedented economic boom and accelerated urbanization over the past few years. To characterize the temporal and spatial distribution and to investigate the effect of various sources on air quality, particularly for Beijing, the

observations of tropospheric SO$_2$ and NO$_2$ vertical column through mobile DOAS are performed from





11 June to 7 July 2013.

Combined with the simultaneous measurement of ground surface concentration through point instruments, the various temporal and spatial distributions of $SO_2$ and $NO_2$ under different wind fields are discussed. For the southwest measurement route, the mean $SO_2$ VCD under southerly wind is 6.09 $\times 10^{16}$ molec./cm$^2$, which is 2.6 times higher than that for north wind ($2.35 \times 10^{16}$ molec./cm$^2$). The near-surface $SO_2$ concentration under southerly wind is 1.24 times higher than that under northerly wind, with the value of 10.78 ppb and 8.69ppb. Except for $SO_2$, the mean $NO_2$ VCD and near-surface $NO_2$ concentration under southerly wind are 1.77 and 1.42 times higher than that under northerly wind. The significant discrepancies of $SO_2$ VCD between the two various wind indicate that the transport from southern NCP area strongly affects the air quality over northern NCP area (like Beijing). And the primary contributors to $SO_2$ along this measurement route are elevated emission sources, like power plant and steel company etc. using the interrelated analysis of VCD and in situ concentration. Moreover, the transport route of the path (Shijiazhuang-Baoding-Beijing) is identified.

However, for the southeast measurement route, we did not find a distinct difference of $SO_2$ VCD under different wind fields, with the value of $3.29 \times 10^{16}$ molec./cm$^2$ and $3.51 \times 10^{16}$ molec./cm$^2$ for south and north wind. The mean near-surface concentration of $SO_2$ for southerly wind is nearly double than that of the value under northerly wind, with the concentration of 23.29 ppb and 11.24 ppb. The $NO_2$ VCDs and near-surface concentrations along the southeast route also do not have significant variations. Under south wind, the $NO_2$ VCD and near-surface concentration are $1.34 \times 10^{16}$ molec./cm$^2$ and 119.12 ppb. Under north wind, the $NO_2$ VCD and near-surface concentration are $9.68 \times 10^{15}$ molec./cm$^2$ and 116.82 ppb. The higher $SO_2$ near-surface concentration along the southeast route indicates the low area sources are the primary contributors to $SO_2$ rather than elevated sources.

Analysis of hot spot shows that the average observed width of air mass is 11.83 km and 17.23 km associated with air mass diffusion. Another interesting finding is that the $NO_2$ near-surface concentration did not significantly enhance for the area of air mass. The lifetime of $NO_2$ is less than that of $SO_2$ and $NO_2$ conversion to other species could account for this unexpected findings. The correlation analysis between $NO_2$ and CO near-surface concentrations shows that $NO_2$ near-surface concentration mainly resulted from vehicle exhaust regardless of specific wind direction.

Furthermore, comparison with OMI $NO_2$ VCDs indicates a reasonable agreement between OMI and mobile DOAS with correlation coefficient ($R^2$) of 0.65. Both datasets have similar spatial patterns. In



most cases, the high level of NO$_2$ VCDs is observed around Shijiazhuang area. However, the fitted slop of 0.55 is significantly less than unity may reflect the existence of some near surface local sources which are insensitive observations or underestimation by OMI. This study will promote the development and extend mobile DOAS technique to rapidly capture the regional distribution of air

pollutants and evaluate the potential transport as well as the use of satellite validation.

*Acknowledgements:* The authors would like to thank Prof. Zhu's group from Peking University for the arrangement of the experiments and providing the data from point instruments during the CAREBEIJING 2013 field campaign. This work was also made possible by the support of the National

Natural Science Foundation of China (41530644 and 41605013), National Key R&D Program (2016YFC0208203 and 2016YFC0201507) and Anhui Province Natural Science Foundation of China (1508085QD71).

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





Table 1 Summary of monitoring information of mobile DOAS. Wind data from airport meteorological data

http://www.wunderground.com

| Cycles | Date | Time(LT) | Routes | Wind Direction | Wind Speed (m/s) |
|--------|------|----------|--------|----------------|------------------|
| **Cycle 1** | 11 June | 10:14-14:00 | BJ-SJZ | BJ: southeast<br>SJZ: southwest | BJ: 2~3<br>SJZ: 1~2 |
| | 12 June | 10:24-14:05 | SJZ-DZ | SJZ: southwest<br>DZ: southwest | SJZ: 1~2<br>DZ: 3~4 |
| | 13 June | 10:20-15:04 | DZ-BD-CZ | DZ: southwest<br>BD: south<br>CZ: southwest | DZ: 4~5<br>BD: 2~3<br>CZ: 4~5 |
| | 14 June | 10:02-13:45 | CZ-ZZ | CZ: southwest<br>ZZ: southwest | CZ: 4~5<br>ZZ: 4~5 |
| | 15 June | 09:57-14:06 | ZZ-BJ | ZZ: south<br>BJ: south | ZZ: 2~3<br>BJ: 2 |
| **Cycle 2** | 17 June | 10:36-14:19 | BJ-SJZ | BJ: northeast<br>SJZ: northeast | BJ: 2~3<br>SJZ: 3 |
| | 18 June | 10:02-13:32 | SJZ-DZ | SJZ: north<br>DZ: north | SJZ: 1~2<br>DZ: 1~2 |
| | 20 June | 10:25-15:05 | DZ-BD-CZ | DZ: northwest<br>BD: northwest<br>CZ: northeast | DZ: 2~3<br>BD: 2~3<br>CZ: 3~4 |
| | 21 June | 09:57-13:24 | CZ-ZZ | CZ: west<br>ZZ: southwest | CZ: 3~4<br>ZZ: 2~3 |
| **Cycle 3** | 24 June | 10:47-14:06 | BJ-SJZ | BJ: southeast<br>SJZ: south | BJ: 2~3<br>SJZ: 2~3 |
| | 25 June | 10:10-14:17 | SJZ-DZ | SJZ: south<br>DZ: south | SJZ: 1~2<br>DZ: 3~4 |
| | 26 June | 09:43-14:01 | DZ-BJ | DZ: southwest<br>BJ: south | DZ: 4~5<br>BJ: 3~4 |
| **Cycle 4** | 2 July | 10:24-14:13 | BJ-SJZ | BJ: northwest<br>SJZ: northwest | BJ: 5~6<br>SJZ: 3~4 |
| | 3 July | 10:26-14:01 | SJZ-DZ | SJZ: southwest<br>DZ: southwest | SJZ: 2<br>DZ: 3~4 |
| | 4 July | 10:12-11:54 | DZ-CZ | DZ: southwest<br>CZ: southeast | DZ: 3~4<br>CZ: 1~2 |
| | 5 July | 09:55-13:40 | CZ-BD-CZ | CZ: northeast<br>BD: northeast | CZ: 3~4<br>BD: 3~4 |
| | 6 July | 09:56-14:23 | CZ-ZZ | CZ: southeast<br>ZZ: southwest | CZ: 2~3<br>ZZ: 2~3 |
| | 7 July | 10:12-13:21 | ZZ-BJ | ZZ: southeast<br>BJ: southeast | ZZ: 2~3<br>BJ: 2~3 |



Table 2 Both results measured through mobile DOAS and point instruments along the southwest and southeast

routes for different wind fields.

| VCD [molec./cm$^2$] | | South Wind | North Wind | Ratio | Average |
|---|---|---|---|---|---|
| **Southwest Route** | SO$_2$ | 6.09×10$^{16}$ | 2.35×10$^{16}$ | 2.69 | 4.22×10$^{16}$ |
| | NO$_2$ | 2.16×10$^{16}$ | 1.22×10$^{16}$ | 1.77 | 1.69×10$^{16}$ |
| **Southeast Route** | SO$_2$ | 3.29×10$^{16}$ | 3.51×10$^{16}$ | 0.94 | 3.40×10$^{16}$ |
| | NO$_2$ | 1.34×10$^{16}$ | 9.68×10$^{15}$ | 1.38 | 1.15×10$^{16}$ |

| Near surface Concentration [ppb] | | South Wind | North Wind | Ratio | Average |
|---|---|---|---|---|---|
| **Southwest Route** | SO$_2$ | 10.78 | 8.69 | 1.24 | 9.74 |
| | NO$_2$ | 130.55 | 92 | 1.42 | 111.28 |
| **Southeast Route** | SO$_2$ | 23.29 | 11.24 | 2.07 | 17.27 |
| | NO$_2$ | 119.12 | 116.82 | 1.02 | 117.97 |

*Ratio: defined as the value under southerly wind/northerly wind

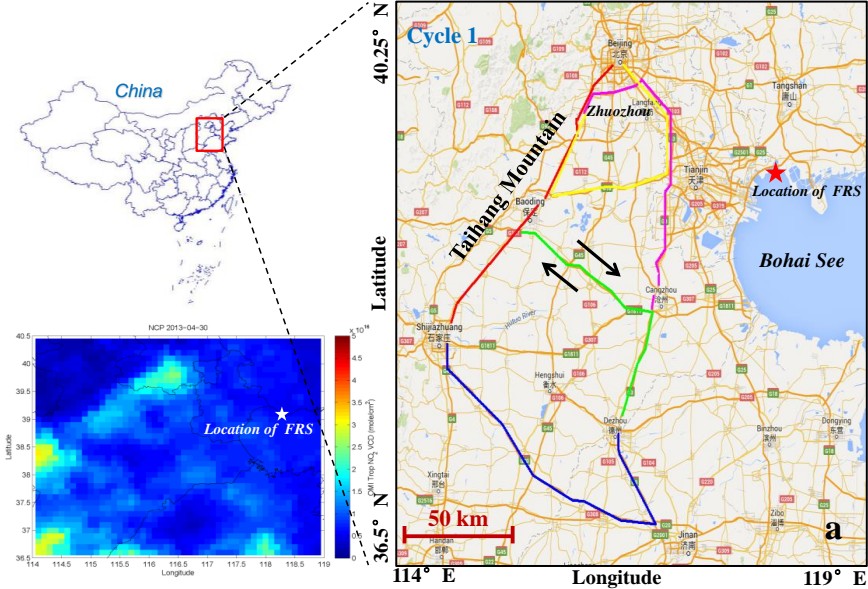





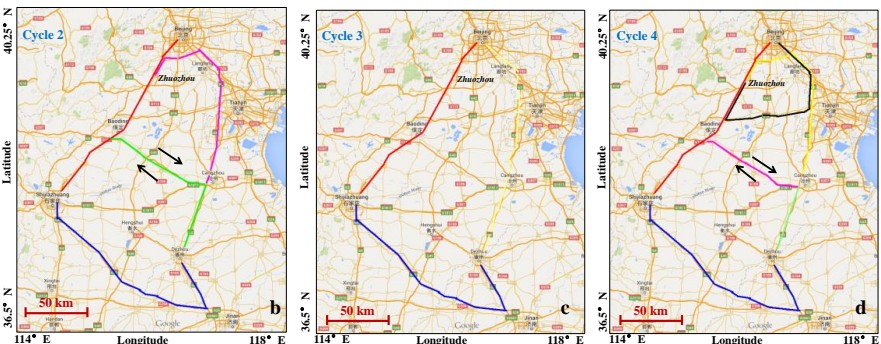

Figure 1: Maps of the mobile measurement areas and routes. The red, blue, green, pink, and yellow tracks show the routes of BJ–SJZ, SJZ–DZ, DZ–BD–CZ, CZ–ZZ, and ZZ–BJ (a). The OMI NO$_2$ VCD on 30 April shows the NO$_2$ VCD of FRS is low (a). (a) also marks the location of FRS, Bohai See, and Taihang Mountain. The red, blue, green, and pink tracks indicate the routes of BJ–SJZ, SJZ–DZ, DZ–BD–CZ, and CZ–ZZ (b). The red, blue, and yellow tracks show the routes of BJ–SJZ, SJZ–DZ, and DZ–BJ (c). The red, blue, green, pink, yellow, and black tracks show the routes of BJ–SJZ, SJZ–DZ, DZ–CZ, CZ–BD–CZ, CZ–ZZ, and ZZ–BJ (d). The black arrows indicate the monitoring route from CZ to BD and return to CZ.

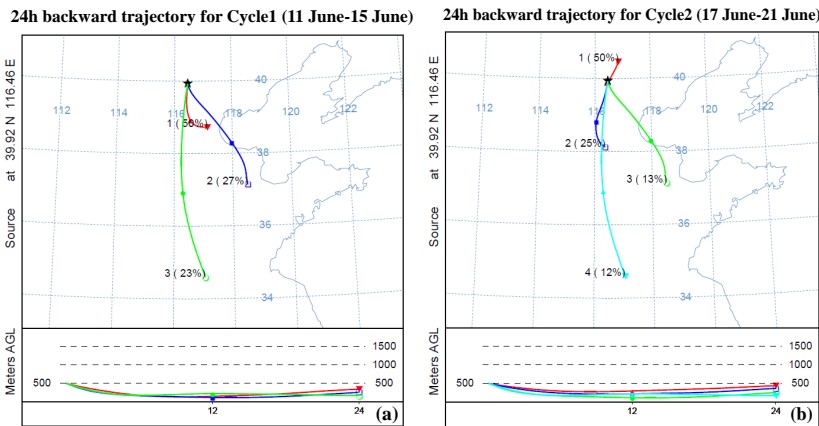



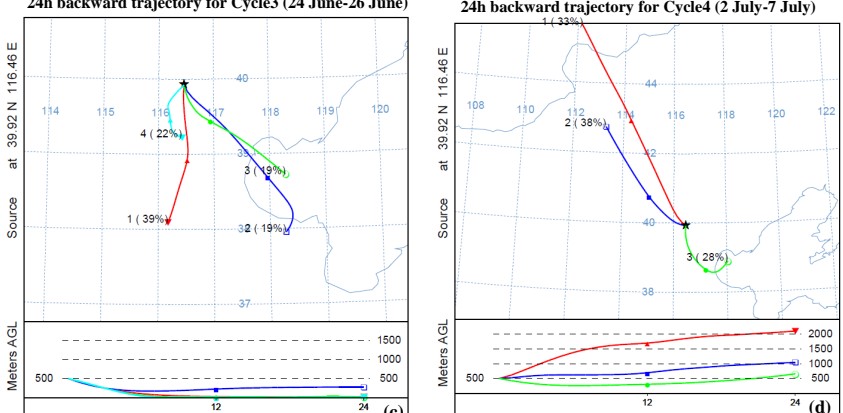

Figure 2: 24 h cluster mean air mass backward trajectories at 500 m height at Beijing for four different cycles. The

black star shows the location of Beijing. The different color lines indicate air mass from different regions. (a), (b),

(c), and (d) show the backward trajectory for Cycles 1, 2, 3, and 4, respectively. The percentages suggest the ratios

5                                                      of air mass in one region.

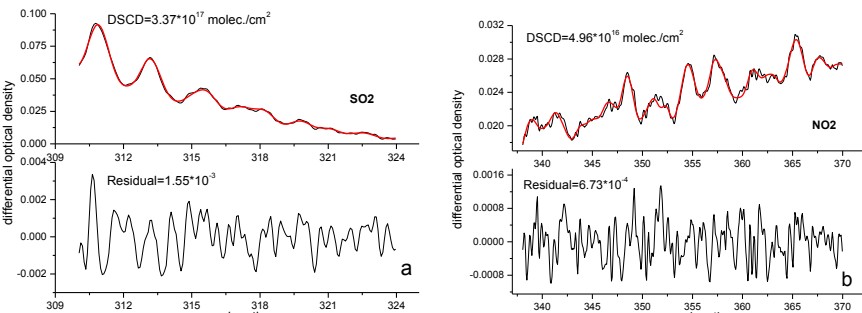

Figure 3: Example of SO$_2$ (a) and NO$_2$ (b) DSCD fit recorded at 13:04 (LT) on 12 June 2013. Black lines denote

the differential optical densities (DODs) of measured spectrum, and red lines show the fit results. The DSCD is the

10                                      SCD (Slant Column Density) with respect to FRS spectrum.



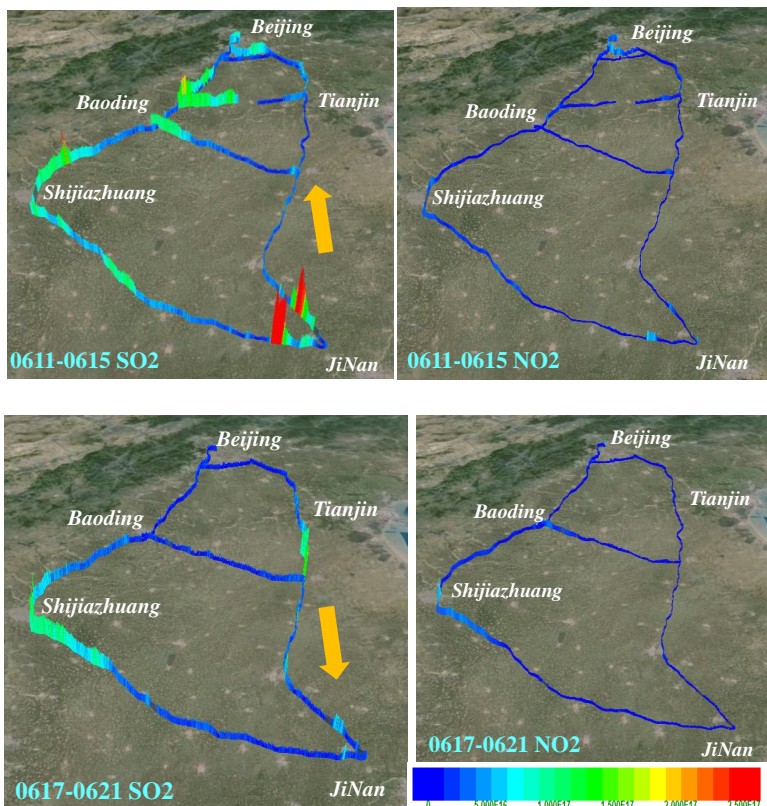

Figure 4: Spatial distributions of SO$_2$ and NO$_2$ VCDs over NCP area for north (17–21 June) and south (11–15 June)

wind fields; the orange arrows show the dominant wind direction.

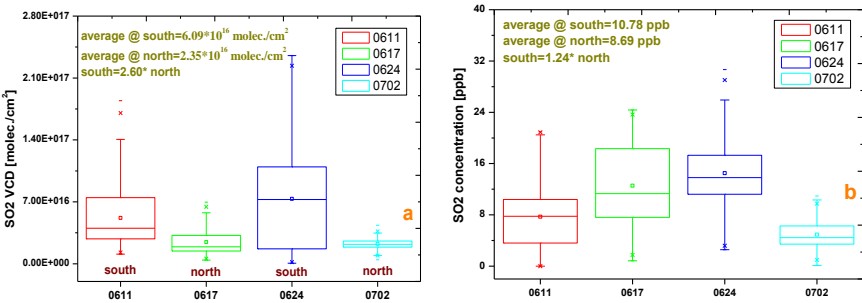

Figure 5: The SO$_2$ VCDs (a) and near-surface concentrations (b) along the southwest route.



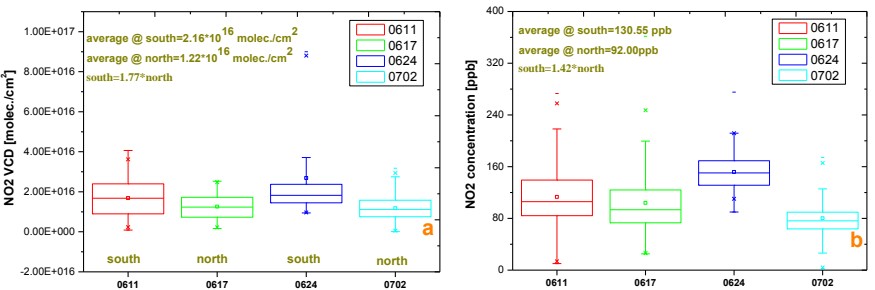

Figure 6: The NO$_2$ VCDs (a) and near-surface concentrations (b) along the southwest route.

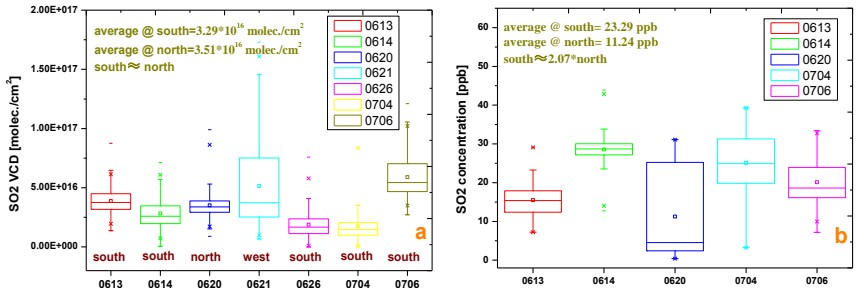

5    Figure 7: The SO$_2$ VCDs (a) and near-surface concentrations (b) along the southeast route; lack of near-surface

data on 21 June and 26 June due to instrument problems.

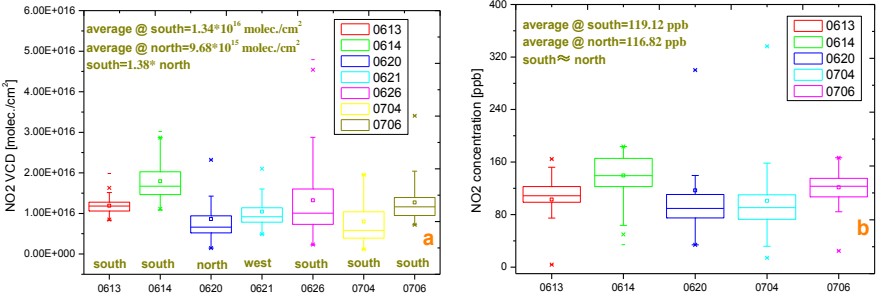

Figure 8: The NO$_2$ VCDs (a) and near-surface concentrations (b) along the southeast route; lack of near-surface

10    data on 21 June and 26 June due to instrument problems.





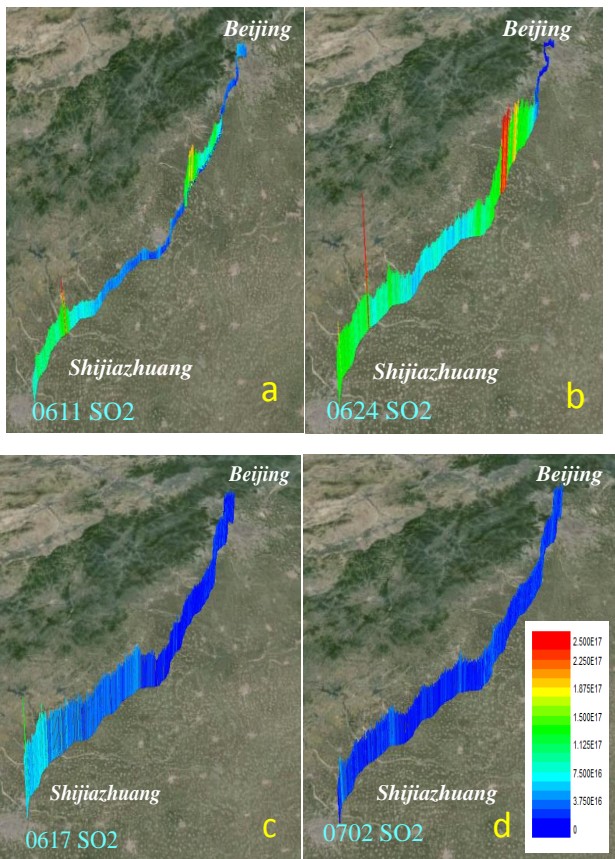

Figure 9: The variations of $SO_2$ VCDs along the southwest measurement routes (Beijing–Shijiazhuang) for south

(a and b) and north (c and d) wind fields.

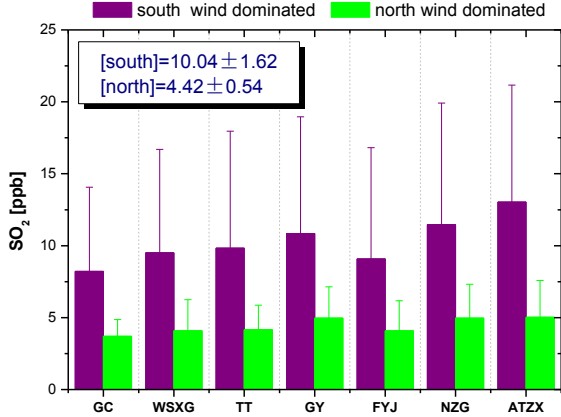

Figure 10: The mean concentrations of $SO_2$ measured at Gucheng (GC), Wanshou Xigong (GSXG), Tian Tan (TT),



Guangyuan (GY), Fuyoujie (FYJ), Nongzhanguan (NZG), and AoTiZhongxin (ATZX) sites based on the south

wind and north wind dominance in Beijing during mobile DOAS observations period. The bars show the standard

deviations of $SO_2$ concentrations.

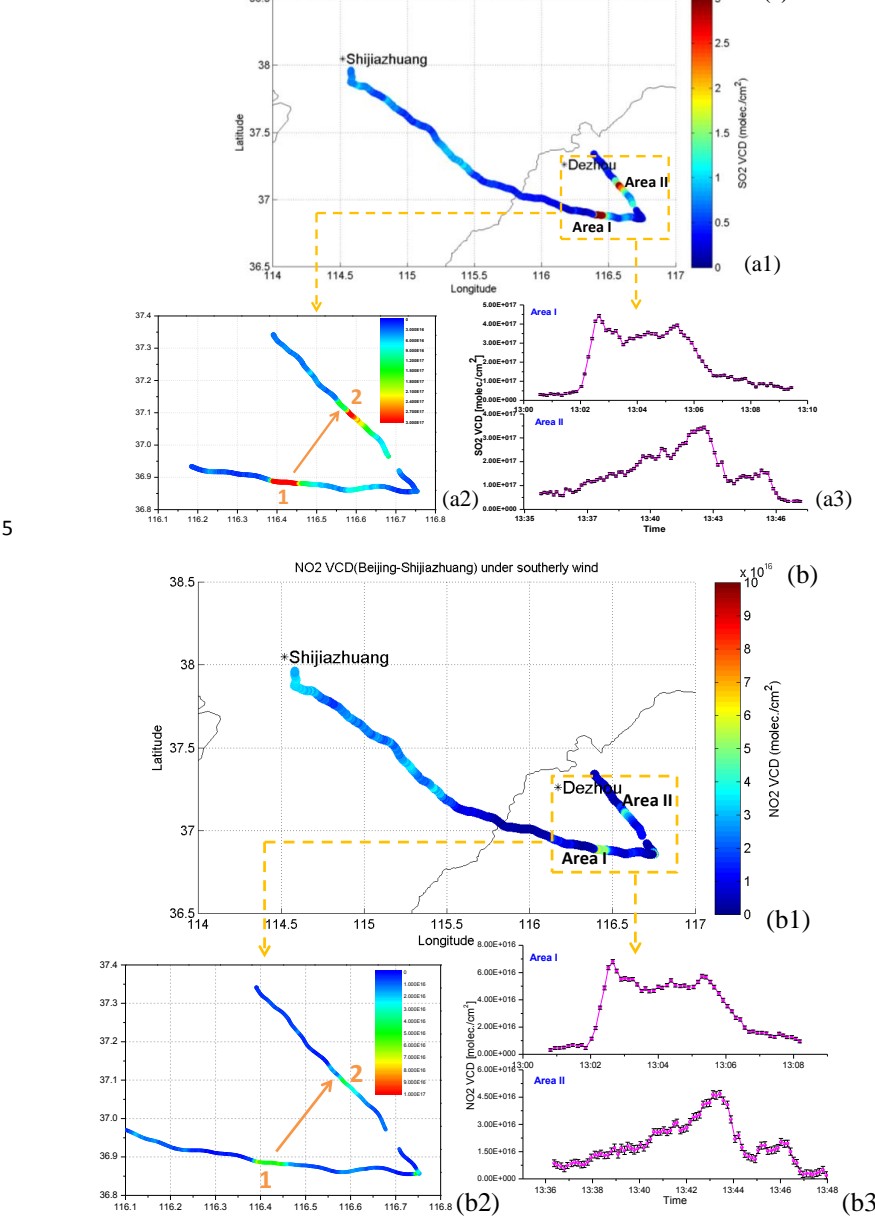



Figure 11: The hot spots of $SO_2$ (a) and $NO_2$ VCDs (b) are observed for the measurement of Shijiazhuang–Dezhou

City under southerly wind on 12 June. (a1): Distribution of $SO_2$ VCDs along the whole measurement route; (a2)

Distribution of $SO_2$ VCDs on hot spot area, where the origin arrow shows the diffusion of air mass from location 1

to location 2; (a3) time series of $SO_2$ VCDs for the polluted air mass for the rectangular area as shown in (a1),

5    where the top figure of (a3) shows the Area I and the bottom for the Area II. (b1), (b2), and (b3) are similar as (a1),

(a2), and (a3), only for $NO_2$.

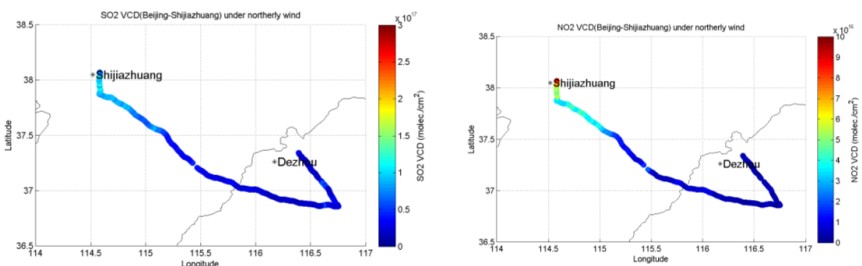

Figure 12: The distributions of $SO_2$ and $NO_2$ VCDs along the Shijiazhuang–Dezhou measurement route for

10                                          northerly wind.



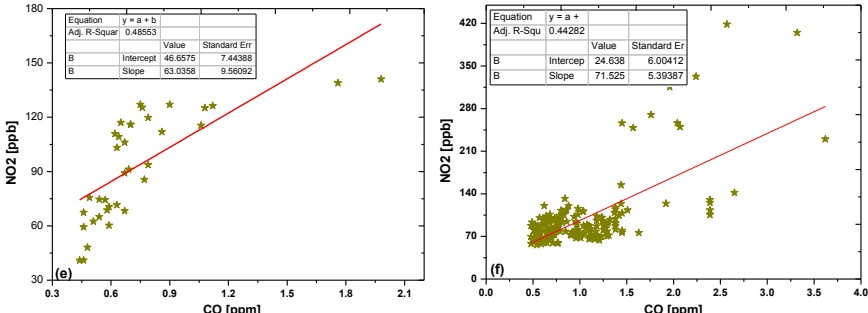

Figure 13: Time series of VCDs and near-surface concentrations of SO$_2$ and NO$_2$ along the route of Shijiazhuang–

Dezhou for south and north wind (a) and (b): SO$_2$ and NO$_2$ VCDs and near-surface concentrations on 3 July under

southerly wind; (c) and (d): SO$_2$ and NO$_2$ VCDs and near-surface concentrations on 18 June under northerly wind;

5     (e): correlation analysis between NO$_2$ and CO near-surface concentrations for south wind (3 July) for the sparse

rectangle area; (f) correlation analysis between NO$_2$ and CO near-surface concentrations for north wind (18 June)

for the sparse rectangle area. The sparse rectangular boxes show polluted air mass area as shown in Fig. 11.

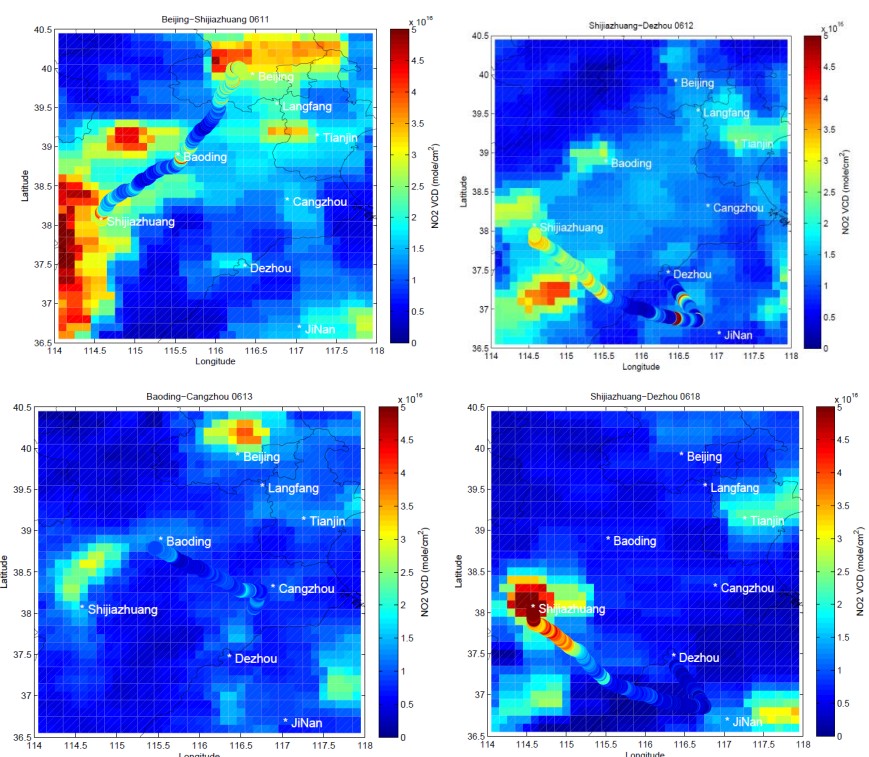




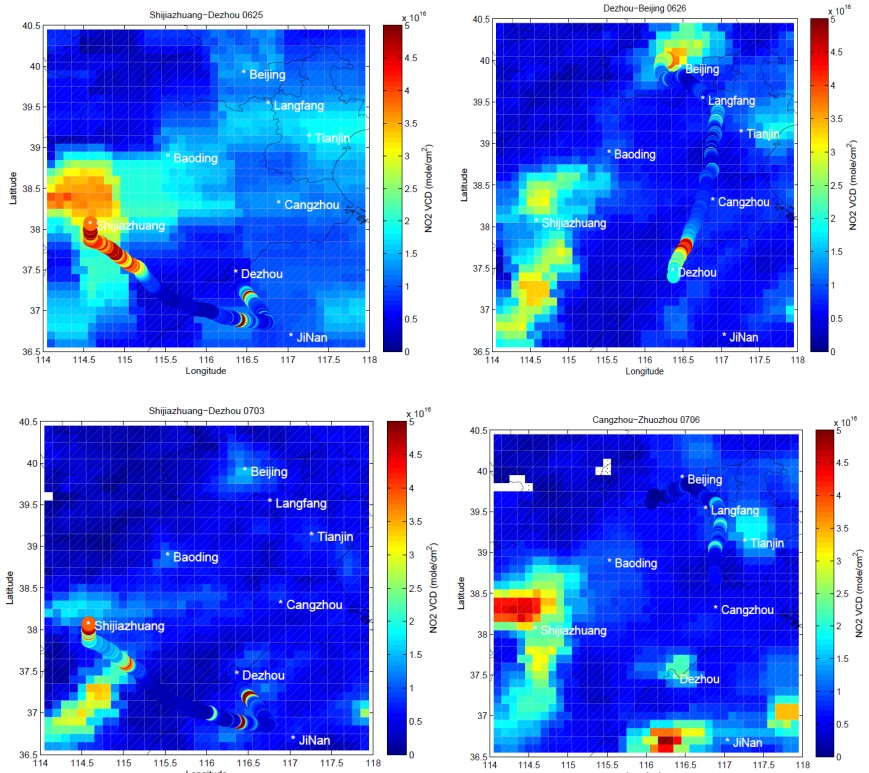

Figure 14: Spatial pattern of NO$_2$ measured through mobile DOAS and OMI. The header of each plot indicates

measured route and date, such as the first plot showing the result of "Beijing–Shijiazhuang" route on 11 June. The

5   color-coded circle indicates the mobile DOAS observations. The grid resolution of OMI was 0.1 °×0.1 °.

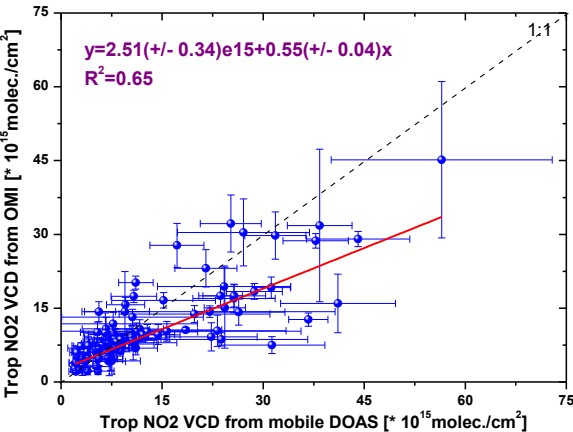

Figure 15: Correlation analysis of mobile DOAS and OMI NO$_2$ VCDs.