# Peer review of "Investigations of Temporal and Spatial Distribution of Precursors SO2 and NO2 Vertical Columns in North China Plain by Mobile DOAS"

_Atmospheric Chemistry and Physics, 2017_

## Referee Comment (RC1) · Anonymous Referee #2 · 26 Oct 2017

This paper gives an overview on mobile DOAS measurements of NO$_2$ and SO$_2$ which were performed in the North China Plain (NCP) in summer 2013. The spatial distributions of the trace gases as well as spatial and temporal variations in these distributions are examined by combining them with wind and in-situ data. Hereby, the effect of pollution transport on the air quality of Beijing is investigated and a transport route for pollution from south to north towards Beijing is identified. Further, a specific emission hot spot is characterised in more detail. Finally, the DOAS dataset is compared to OMI data, where a reasonable agreement is found considering the different measurement strategies. Since the topic air pollution in China (especially in the major cities such as Beijing) is of high importance and reasonable results and conclusions are obtained the

paper is suited for publication in ACP. Nevertheless, some points need to be clarified and commented by the authors before final publication.

General Comments: The English language is in general good and fluent. However, the use of articles and prepositions has to be improved as well as the use of plural/singular forms. Additionally, some sentences are really hard to understand (for more details see specific comments).

Further, I personally don't like the term "point measurements/instrument" (e.g. page 4, line 11 or 15 / Page 5, line 25) which is used throughout the paper. Since you are referring to in-situ instruments here, I would suggest to use the term "in-situ measurements/instrument" in the paper.

Major Comments:

**Section 2.4:** Since the DOAS principle is the major measurement method of this paper I have several comments on this section:

Page 6 starting from line 20: Here several sentences are really hard to understand (since they are very long). I suggest that you rephrase the whole part.

Page 7, line 1-8: Here, I suggest to summarise all the fit settings in a table (one for $NO_2$ and one for $SO_2$) and to also give all the references for all used cross-section spectra. Is there a specific reason why the Greenblatt et al. cross-section is used (and not the newer Thalman et al.)? I would suggest to test if there is a significant effect on the trace gas results when using the newer Thalman et al. cross-section. Further, which high-resolution solar spectrum was used?

Perhaps you could give a reference for equation (1) and I think it should be $DSCD_{trop}$ in the last part of the equation.

The estimations of the errors is quite optimistic, but reasonable. However, I wonder how clouds are treaded in calculation of the AMFs.

So in general I ask you to revise the whole section 2.4. considering my comments.

**Page 8, line 13-16:** I agree that atmospheric chemistry is much too complex to discuss everything in detail here. Further, I agree that the wind direction has the main influence here. However, it would be interesting if you could comment also on the other parameters: temperature, humidity and pressure. For temperature you already did this quickly in section 2.1. Perhaps similar statements could be given for the other parameters, At least in your comment.

**Section 3.1.3:** In general I agree that your measurements indicate that the air quality in Beijing is negatively affected under southerly wind. However, I wonder were exactly the emission sources are located (and I think this is one of the main questions you are trying to answer in this section and the paper). At the end of section 3.1.1 you state that the high $SO_2$ VCDs under southerly wind along the Taihang Mountain are caused by emission in the south/southwest of the measurement region which sounds reasonable here and I agree to that. But on the other side you state (in section 3.1.3, e.g. line 30, 31) that the large emission sources (of $SO_2$) are located along the southwest route/region. The latter might explain the higher $SO_2$ VCDs along the southeastern route under westerly wind (such as on the 21st of June). This would mean that the emission sources of SO2 are located along the southwest route which would first contradict to the fact that there is a certain dependence of the VCDs in the southwest on the wind direction (north vs. south). Second, I would then expect that along the southwest route we should always see high $SO_2$ independent from the wind direction. I think it is quite obvious that the main emission sources are outside the cycles. Could you clarify this or comment on this? In general I think it would help to understand everything in more detail if you would show plots as in figure 4 for all cycles (perhaps as appendix).

**Page 13, line 8-10 + Figure 13 e and f:** First, I suggest to split up this figure (Figure 13) and give an own number for Figure 13 e and f together. Further, I wonder why you are only showing a correlation analysis for two specific days. I think you measured this

route (SJZ to DZ) four times. So you should bin your four data sets with respect to wind direction and then do a correlation analysis for the complete dataset which would be more robust.

Minor/Specific Comments:

Page 3, line 5-9: perhaps give a rough (relative) number for the "large proportions in particulate matter" and the "significant reduction"

Page 3, line 9, 10: I think the term "$NO_x$" should be introduced and specified first, since beforehand only $NO_2$ is mentioned

Page 3, line 22, 23: I would rephrase as follows: "...based on measurements using a mobile laboratory, ..."

Page 3, line 25: I would rephrase as follows: "Model simulations, another method, can obtain distributions, ... and major transport channels of ..."

Page 3, line 29: Since there have been several publications dealing with mobile (MAX-) DOAS measurements I would skip "novel" here.

Page 4, line 4, 5: Shaiganfar et al. published a paper in 2015 (*New concepts for the comparison of tropospheric $NO_2$ column densities derived from car-MAX-DOAS observations, OMI satellite observations and the regional model CHIMERE during two MEGAPOLI campaigns in Paris 2009/10, AMT, 2015*) I think it should be additionally mentioned here.

Page 4 , line 22: add "column" to "...vertical *column* densities..."

Page 5, line 21, 22: Here I think it is not of large importance which car is used and which dimensions it has. So I would just state that the instrument is mounted on a car/ mobile laboratory.

Page 6, line 6: Please add the year to the citation.

Section 2.3: Here I wonder how representative the 500 m simulated trajectories are for the ground-based wind data. As already discussed by you the wind data has some contradictions as for example for cycle 4. Could you briefly comment on this?

Page 8, line 9: Some words are missing in this sentence.

Page 9, line 5: Perhaps the "city effect" should be mentioned as ". . .affected by local emissions within the cities. . ." or something like that

Section 3.1.1: In general this section refers to Figure 4. Two comments on this figure: first, the dates in the left corners of the subplots are really hard to read. Second, I would suggest to use different colour scales for NO2 and SO2. This would more pronounced the elevated VCDs around the cities.

Further the city of Ji'nan is written in three different ways throughout the paper perhaps this should be harmonised.

Page 11, line 26: I would say ". . .they are less pronounced. . .", since the hot spots are still visible (see figure 4c).

Page 12, line 16: Is there any hint which kind of source this might be? Or is it the city itself?

Page 12, line 18: Please add which day is shown in figure 12 since it somehow contradicts to figure 4 where the hot spots are still visible but less pronounced.

Page 13, line 16: please change to something like ". . .to measure spectra of scattered sunlight. . .".

Page 13, line 18: "The data *analysis* consisted of three steps . . . tropospheric $NO_2$ VCD*s*"

Page 13, line 25: here the word "grid" is missing.

Section 3.3: Especially in this section the English language needs improvement. Further, which error is exactly represented by the error bars in the correlation plot? Are the errors taken into account when performing the linear regression? This should be added to the text.
* * *

---

## Referee Comment (RC2) · Anonymous Referee #3 · 21 Nov 2017

This paper gives an overview on mobile DOAS measurements of precursors SO2 and NO2 vertical columns in NCP in summer of 2013. The different temporal and spatial distributions of SO2 and NO2 vertical column density (VCD) over this area are characterized by combining them with wind and in-situ data. The transport route and emission sources are identified using the interrelated analysis between in situ and mobile DOAS observations. And also a specific hot spot is characterized in more detail. Finally, a reasonable agreement exists between OMI and mobile DOAS observations with correlation coefficient (R2) of 0.65 for NO2 VCDs. I think this paper presents a useful data set, and a good insightful analysis. The present data of this work are reasonable published in ACP. However, following suggestions need to be considered before final

publication.

1. Some expression should be consistent throughout the paper, like point instrument data and in-situ data, JiNan and Ji'nan... 2. Maybe there are some mistakes of titles in Figure 11 and Figure 12. Please correct them. 3. Section 2.4, I suggest to list all the fit settings in a table for NO2 and SO2. 4. Section 3.1.1: I agree that the wind direction has the main influence on air mass variations. However, you could also give other parameters: humidity and pressure, as discussion in section 2.1 about temperature comment. 5. The quality of figure 4 should be improved, like dates, color bar. I think it is best to give all results in Figure 4 for all cycles. You can present them in supplement material. 6. Figure 9: the dates in the left corners of the subplots are really hard to read. Please correct them. 7. Figure 13 e and f: I suggest to give and another number of figure 13 e and f together. And I think if you make a correlation analysis using all data regarding to wind direction, it is more robust. 8. Figure 15: could you tell us which error is exactly represented by the error bars, standard deviations? Please clarify it.

Please also note the supplement to this comment:
https://www.atmos-chem-phys-discuss.net/acp-2017-719/acp-2017-719-RC2-supplement.pdf
* * *
This paper gives an overview on mobile DOAS measurements of precursors SO₂ and NO₂ vertical columns in NCP in summer of 2013. The different temporal and spatial distributions of SO₂ and NO₂ vertical column density (VCD) over this area are characterized by combining them with wind and in-situ data. The transport route and emission sources are identified using the interrelated analysis between in situ and mobile DOAS observations. And also a specific hot spot is characterized in more detail. Finally, a reasonable agreement exists between OMI and mobile DOAS observations with correlation coefficient ($R^2$) of 0.65 for NO₂ VCDs. I think this paper presents a useful data set, and a good insightful analysis. The present data of this work are reasonable published in ACP. However, following suggestions need to be considered before final publication.

1. Some expression should be consistent throughout the paper, like point instrument data and in-situ data, JiNan and Ji'nan…

2. Maybe there are some mistakes of titles in Figure 11 and Figure 12. Please correct them.

3. Section 2.4, I suggest to list all the fit settings in a table for NO₂ and SO₂.

4. Section 3.1.1: I agree that the wind direction has the main influence on air mass variations. However, you could also give other parameters: humidity and pressure, as discussion in section 2.1 about temperature comment.

5. The quality of figure 4 should be improved, like dates, color bar. I think it is best to give all results in Figure 4 for all cycles. You can present them in supplement material.

6. Figure 9: the dates in the left corners of the subplots are really hard to read. Please correct them.

7. Figure 13 e and f: I suggest to give and another number of figure 13 e and f together. And I think if you make a correlation analysis using all data regarding to wind

Fig. 1.

---

## Author Comment (AC1) · 11 Dec 2017

We would like to thank the referee for taking the time to read the paper carefully and provide helpful suggestions to improve the paper. We have revised the paper according to the referee's comments carefully, where the revised parts are indicated by red font. The detailed revisions are described as follows: Referee #2 General comments: 1. The English language is in general good and fluent. However, the use of articles and prepositions has to be improved as well as the use of plural/singular forms. Additionally, some sentences are really hard to understand (for more details see specific comments). Response: We have tried our best to improve the English language

throughout the paper, e.g. we have corrected the usage of prepositions, plural/singular forms etc. If it is not still meet the quality required by the journal, we think we can get help from language editorial service offered by the journal. 2. Further, I personally don't like the term "point measurements/instrument" (e.g. page 4, line 11 or 15 / Page 5, line 25) which is used throughout the paper. Since you are referring to in-situ instruments here, I would suggest to use the term "in-situ measurements/instrument" in the paper. Response: We have changed the term "point measurements/instrument" to "in-situ measurements/instrument" in the paper.

Major comments: Section 2.4: Since the DOAS principle is the major measurement method of this paper. I have several comments on this section: 3. Page 6 starting from line 20: Here several sentences are really hard to understand (since they are very long). I suggest that you rephrase the whole part. Response: We have divided some long sentences into some short sentences and rephrased the whole part, line 25-30, Page 6, line 1-3, Page 7. 4. Page 7, line 1-8: Here, I suggest to summarise all the fit settings in a table (one for NO2 and one for SO2) and to also give all the references for all used cross-section spectra. Is there a specific reason why the Greenblatt et al. cross-section is used (and not the newer Thalman et al.)? I would suggest to test if there is a significant effect on the trace gas results when using the newer Thalman et al. cross-section. Further, which high-resolution solar spectrum was used? Response: We have summarized all the fit settings in Table 2 and give all the references for all used cross-section spectra. In addition, we adapted from experiences during MAD-CAT (http://joseba.mpch-mainz.mpg.de/mad_cat.htm) intercomparison campaign and changed some cross-section for NO2 fitting. One clerical error occurred in the fitting window of NO2. Actually, it is 338nm-370nm for NO2 fitting in this study, not 338nm-379nm. We have corrected it. The usage of the Greenblatt et al. cross-section is just from our previous experiences (Wu et al., 2013, AMT), no specific reason. Also we have tested the effect of cross-section on the result based on Windoas retrieval. It is found that the difference is lower than 1% for three different scenarios e.g. high NO2 VCD, middle NO2 VCD and low NO2 VCD. So, we think the effect is not significant

compared with other error source and we ignore it in this study. The high-resolution solar spectrum from Kurucz et al., 1984 is used. These revised parts are indicated by red font in Sect. 2.4. 5. Perhaps you could give a reference for equation (1) and I think it should be DSCDtrop in the last part of the equation. Response: We have added the reference for equation (1) and corrected the last part of the equation, Line 17-18, Page 7. 6. The estimations of the errors is quite optimistic, but reasonable. However, I wonder how clouds are treaded in calculation of the AMFs. Response: Of course, the clouds have a significant effect on results retrieved by remote sensing observations. Clouds effect is important and difficult in the retrieval of trace gas results. In this study, we treaded the clouds as following two aspects: 1. we carried out observations under the condition of less cloud cover as soon as possible; 2. we regard the clouds as aerosol in the radiative transfer model for calculation of the AMFs to reduce the effect of clouds for cloudy day. 7. Page 8, line 13-16: I agree that atmospheric chemistry is much too complex to discuss everything in detail here. Further, I agree that the wind direction has the main influence here. However, it would be interesting if you could comment also on the other parameters: temperature, humidity and pressure. For temperature you already did this quickly in section 2.1. Perhaps similar statements could be given for the other parameters, At least in your comment. Response: The humidity and pressure are in the range of 32% ∼61% and 994hPa∼1009hPa during the entire measurement period in NCP area. Similar statements of humidity and pressure have been given in the paper, line 10-12, Page 5. 8. In general I agree that your measurements indicate that the air quality in Beijing is negatively affected under southerly wind. However, I wonder were exactly the emission sources are located (and I think this is one of the main questions you are trying to answer in this section and the paper). At the end of section 3.1.1 you state that the high $SO_2$ VCDs under southerly wind along the Taihang Mountain are caused by emission in the south/southwest of the measurement region which sounds reasonable here and I agree to that. But on the other side you state (in section 3.1.3, e.g. line 30, 31) that the large emission sources (of $SO_2$) are located along the southwest route/region. The latter might explain the higher $SO_2$

VCDs along the southeastern route under westerly wind (such as on the 21st of June). This would mean that the emission sources of SO2 are located along the southwest route which would first contradict to the fact that there is a certain dependence of the VCDs in the southwest on the wind direction (north vs. south). Second, I would then expect that along the southwest route we should always see high SO2 independent from the wind direction. I think it is quite obvious that the main emission sources are outside the cycles. Could you clarify this or comment on this? In general I think it would help to understand everything in more detail if you would show plots as in figure 4 for all cycles (perhaps as appendix). Response: We have further analyzed the different spatial distributions under different wind fields. And give the locations of main SO2 sources in the southwest of measurement region as shown in Fig. 4. We think the main source S1 and S2 in the measurement route, which can explain the SO2 high VCDs in Shijiazhuang for north wind. However, the important SO2 sources S3 located outside the cycles, which maybe large area sources, not just a point source. This can result in the SO2 VCDs have a significant increase in the southwest of measurement region when air mass come from south. We also have further discussed the wind field on 21 June. The 24h backward trajectory of 500m show that the air mass come from west/southwest direction near mobile DOAS measurement time, but from northeast direction when the time moves forward longer in Cangzhou (the location of peak value) city as shown in Fig. S2. We think the high SO2 VCDs on 21 June maybe caused by local emission and transport from northeast direction. These revised parts are indicated by red font in Sect. 3.1.3. In addition, we have given all results in Fig. 4 and Fig. S1. Fig. 4 show the results for Cycle 1 and 2. Fig. S1 show the results for Cycle 3 and 4.

9. Page 13, line 8-10 + Figure 13 e and f: First, I suggest to split up this figure (Figure 13) and give an own number for Figure 13 e and f together. Further, I wonder why you are only showing a correlation analysis for two specific days. I think you measured this route (SJZ to DZ) four times. So you should bin your four data sets with respect to wind direction and then do a correlation analysis for the complete dataset which would be more robust. Response: We have split up the Fig. 13 and given an own number for

Fig. 13 e and f. Now, these results are shown in Fig. 13 and Fig. 14. There are four times (12 June, 18 June, 25 June and 3 July) to measure the route (SJZ to DZ). Unfortunately, the in-situ instruments have some problems and lack of in-situ data on 25 June. So, total time of measurements of SJZ-DZ route is three, 12 June (south wind), 18 June (north wind) and 3 July (south wind). We have binned three data sets with respect to wind direction and do a correlation analysis for the complete dataset. The results show that NO2 near-surface concentration mainly results from vehicle exhaust, although the correlation coefficient under southerly wind slightly better than that under northerly wind during the measurement periods as shown in line 21-24, Page 13.

Minor/Specific comments: 1. Page 3, line 5-9: perhaps give a rough (relative) number for the "large proportions in particulate matter" and the "significant reduction" Response: We have added them in the text, line 4 and 7, Page 3. 2. Page 3, line 9, 10: I think the term "NOx" should be introduced and specified first, since beforehand only NO2 is mentioned Response: We have added it in the text, line 9, Page 3. 3. Page 3, line 22, 23: I would rephrase as follows: ". . .based on measurements using a mobile laboratory, ..." Response: We have rephrased them, line 21-22, Page 3. 4. Page 3, line 25: I would rephrase as follows: "Model simulations, another method, can obtain distributions, ... and major transport channels of . . ." Response: We have rephrased them, line 24-25, Page 3. 5. Page 3, line 29: Since there have been several publications dealing with mobile (MAX-) DOAS measurements I would skip "novel" here. Response: We have deleted the "novel" here, line 28, Page 3. 6. Page 4, line 4, 5: Shaiganfar et al. published a paper in 2015 (New concepts for the comparison of tropospheric NO2 column densities derived from car-MAX-DOAS observations, OMI satellite observations and the regional model CHIMERE during two MEGAPOLI campaigns in Paris 2009/10, AMT, 2015) I think it should be additionally mentioned here. Response: We have cited the reference here, line 4, Page 4. 7. Page 4 , line 22: add "column" to ". . .vertical column densities. . ." Response: We have added the "column" to ". . .vertical column densities. . .", line 20, Page 4. 8. Page 5, line 21, 22: Here I think it is not of large importance which car is used and which dimensions it has. So I

would just state that the instrument is mounted on a car/mobile laboratory. Response: We have corrected them, line 20-21, Page 5. 9. Page 6, line 6: Please add the year to the citation. Response: We have added it, line 5, Page 6. 10. Section 2.3: Here I wonder how representative the 500 m simulated trajectories are for the ground-based wind data. As already discussed by you the wind data has some contradictions as for example for cycle 4. Could you briefly comment on this? Response: We think there are no contradictions between Table 1 and Fig. 2 for Cycle 4. It seems to be that the wind dominated by north from Fig. 2 and dominated by south in Table 1 for Cycle 4. It feels like there are some contradictions. However, two main reasons can account for the differences. Firstly, the backward trajectories are simulated at Beijing site and this ratio represents Beijing area. Secondly, the ratio results from calculations of 60 trajectories in five days for Cycle 4. However, the wind data described in Table 1mainly focuses on the time of mobile DOAS observations, line 17-23, Page 6. 11. Page 8, line 9: Some words are missing in this sentence. Response: We have rephrased the sentence, line 11, Page 8. 12. Page 9, line 5: Perhaps the "city effect" should be mentioned as ". . .affected by local emissions within the cities. . ." or something like that Response: We have corrected it, line 8, Page 9. 13. Section 3.1.1: In general this section refers to Figure 4. Two comments on this figure: first, the dates in the left corners of the subplots are really hard to read. Second, I would suggest to use different colour scales for NO2 and SO2. This would more pronounced the elevated VCDs around the cities. Response: We have improved the Fig. 4. 14. Further the city of Ji'nan is written in three different ways throughout the paper perhaps this should be harmonised. Response: We have harmonized them throughout the paper. The city of Ji'nan is written as JiNan now. 15. Page 11, line 26: I would say ". . .they are less pronounced. . .", since the hot spots are still visible (see figure 4c). Response: We have corrected it, line 9, Page 12. 16. Page 12, line 16: Is there any hint which kind of source this might be? Or is it the city itself? Response: We think it is a larger area emission source from Liaocheng direction, maybe is the city itself. 17. Page 12, line 18: Please add which day is shown in figure 12 since it somehow contradicts to figure 4 where the hot

spots are still visible but less pronounced. Response: Fig. 12 presents the data on 18 June. Actually, the expression of "less pronounced" is more exact than the expression of "not found". We have corrected it, line 1, Page 13. 18. Page 13, line 16: please change to something like "...to measure spectra of scattered sunlight...". Response: We have corrected it, line 29-30, Page 13. 19. Page 13, line 18: "The data analysis consisted of three steps ...tropospheric NO2 VCDs" Response: We have corrected it, line 2, Page 14. 20. Page 13, line 25: here the word "grid" is missing Response: We have added it, line 9, Page 14. 21. Section 3.3: Especially in this section the English language needs improvement. Further, which error is exactly represented by the error bars in the correlation plot? Are the errors taken into account when performing the linear regression? This should be added to the text. Response: We have tried our best to improve the English language in this section. The error bars indicate the OMI error and the standard deviation of mobile DOAS observations within $0.1° \times 0.1°$ pixel. They are also taken into account when performing the linear regression, line 22-24, Page 14.

Thanks for your opinions and very appreciated your time. If you have any questions about the manuscripts, please let me know.

Please also note the supplement to this comment:
https://www.atmos-chem-phys-discuss.net/acp-2017-719/acp-2017-719-AC1-supplement.pdf

---

## Author Comment (AC2) · 11 Dec 2017

We would like to thank the referee for taking the time to read the paper carefully and provide helpful suggestions to improve the paper. We have revised the paper according to the referee's comments carefully, where the revised parts are indicated by red font. The detailed revisions are described as follows:

Referee #3 1. Some expression should be consistent throughout the paper, like point instrument data and in-situ data, JiNan and Ji'nan. . . Response: We have corrected them. The point instrument/data and Ji'nan have been written as in-situ instrument/data and JiNan, respectively. 2. Maybe there are some mistakes of titles in Figure

[Figure]

11 and Figure 12. Please correct them. Response: We have corrected them as shown in Fig. 11 and Fig. 12. 3. Section 2.4, I suggest to list all the fit settings in a table for NO2 and SO2. Response: The all fit settings for NO2 and SO2 are summarized in Table 2. 4. Section 3.1.1: I agree that the wind direction has the main influence on air mass variations. However, you could also give other parameters: humidity and pressure, as discussion in section 2.1 about temperature comment. Response: The humidity and pressure are in the range of 32% ∼61% and 994hPa∼1009hPa during the entire measurement period in NCP area. Similar statements of humidity and pressure have been given in the paper, line 10-12, Page 5. 5. The quality of figure 4 should be improved, like dates, color bar. I think it is best to give all results in Figure 4 for all cycles. You can present them in supplement material. Response: Fig. 4 has been improved and all results are presented. The results for Cycle 3 and Cycle 4 are demonstrated in Fig .S1 in supplement material. 6. Figure 9: the dates in the left corners of the subplots are really hard to read. Please correct them. Response: We have corrected the Fig. 9. 7. Figure 13 e and f: I suggest to give and another number of figure 13 e and f together. And I think if you make a correlation analysis using all data regarding to wind direction, it is more robust. Response: We have split up the Fig. 13 and given an own number for Fig. 13 e and f. Now, these results are shown in Fig. 13 and Fig. 14. There are four times (12 June, 18 June, 25 June and 3 July) to measure the route (SJZ to DZ). Unfortunately, the in-situ instruments have some problems and lack of in-situ data on 25 June. So, total time of measurements of SJZ-DZ route is three, 12 June (south wind), 18 June (north wind) and 3 July (south wind). We have binned three data sets with respect to wind direction and do a correlation analysis for the complete dataset. The results show that NO2 near-surface concentration mainly results from vehicle exhaust, although the correlation coefficient under southerly wind slightly better than that under northerly wind during the measurement periods as shown in line 21-24, Page 13. 8. Figure 15: could you tell us which error is exactly represented by the error bars, standard deviations? Please clarify it. Response: The error bars indicate the OMI error and the standard deviation of mobile DOAS observations within $0.1°\times0.1°$

pixel, line 22-24, Page 14.

Thanks for your opinions and very appreciated your time. If you have any questions about the manuscripts, please let me know.

Please also note the supplement to this comment:
https://www.atmos-chem-phys-discuss.net/acp-2017-719/acp-2017-719-AC2-supplement.pdf

---

## Author Response (AR2)

**Response to Referee's Comments**

We would like to thank the referee for taking the time to read the paper again and provide helpful suggestions to improve the paper. We have revised the paper according to the referee's comments carefully, where the revised parts are indicated by blue font. The detailed revisions are described as follows:

**Some last comments/ questions :**

1. In general there are still some grammatical and language mistakes, however, all in all it has improved. Perhaps the language editorial service offered by the journal might solve this as you mentioned in your response.

**Response:** We have improved some grammatical and language mistakes. If it is not still meet the quality required by the journal, we can get help from language editorial service offered by the journal.

2. Starting on page 6 last line until page 7 line 2, for me this sentence is really hard to understand. Perhaps you could rephrase this one.

**Response:** We have rephrased it, line 30, Page 6 and line 1-4, Page 7.

3. Page 7, line 2-3: I would write: "Fit examples for $NO_2$ and $SO_2$ are illustrated in Fig. 3."

**Response:** We have corrected it, line 4, Page 7.

4. In the first version of you manuscript you used the Greenblatt et al., 1990 O4 cross-section. You argued that the error is around 1 % and therefore negligible, however, you are now using the Thalman et al. cross-section. Is this correct?

**Response:** Maybe we have not express clearly in the first response. We want to tell that we have tested the effects of the $O_4$ cross sections from Greenblatt et al., 1990 and the Thalman et al. cross-section on the $NO_2$ results. It is found that the difference of $NO_2$ value based on usage of the two different $O_4$ cross sections is  around 1% for three different scenarios e.g. high $NO_2$ VCD, middle $NO_2$ VCD and low $NO_2$ VCD. Compared with other error source, we ignore the difference in this study. However, in order to follow the latest fit settings, we now change the newer one, the Thalman $O_4$ cross-section.

5. I think in Table 2 something went wrong. I guess not all the cross-sections are from Bogumil et al., 2003. Please correct this in the final version.

**Response:** I have checked the fit settings for $SO_2$ analysis carefully again. Thank you for reading carefully and it really has something wrong in $NO_2$ and HCHO cross sections in Table 2. We have now corrected them in Table 2.

6. One further point: on page 7, line 17 you cite Hönninger et al., 2004, here (and also in the bibliography) you missed the two dots on the "o", this should be corrected in the final version.

**Response:** Because the "ö" is a German character. We could not type it from my computer. But now we copy this character from above sentence into the manuscript, line 18, Page 7 and line 12, Page 18.

7. Page 8, lines 13 – 16: here you could give a short cross reference to section 2.1

**Response:** We have corrected it, line 17, Page 8.

Thanks for your opinions and very appreciated your time again.

If you have any questions about the manuscripts, please let me know.